# Change of Direction Speed and Technique Modification Training Improves 180° Turning Performance, Kinetics, and Kinematics

**DOI:** 10.3390/sports9060073

**Published:** 2021-05-24

**Authors:** Thomas Dos’Santos, Christopher Thomas, Alistair McBurnie, Paul Comfort, Paul A. Jones

**Affiliations:** 1Musculoskeletal Science and Sports Medicine Research Centre, Department of Sport and Exercise Sciences, Manchester Metropolitan University, Manchester M15 6BH, UK; 2Directorate of Sport and Psychology, University of Salford, Greater Manchester M6 6PU, UK; c.thomas2@edu.salford.ac.uk (C.T.); p.a.jones@salford.ac.uk (P.A.J.); 3Department of Football Medicine and Science, Manchester United Football Club, AON Training Complex, Manchester M31 4BH, UK; alistairjmcburnie@gmail.com (A.M.); p.comfort@salford.ac.uk (P.C.)

**Keywords:** intervention, braking, technique feedback, deceleration, external cues, pivot

## Abstract

This study aimed to examine the effects of change of direction (COD) speed and technique modification training on 180° turning performance (completion time, ground contact time [GCT], and exit velocity), kinetics, and kinematics. A non-randomised 6 week intervention study was administered. Thirteen male multidirectional sport athletes formed the intervention group (IG), participating in two COD speed and technique modification sessions per week. A total of 12 male multidirectional sport athletes formed the control group (CG). All subjects performed six modified 505 trials, whereby pre-to-post-intervention biomechanical changes were evaluated using three-dimensional motion analysis. Two-way mixed analysis of variances revealed significant interaction effects (group × time) for completion time, mean horizontal propulsive force (HPF), horizontal to vertical mean braking and propulsive force ratios for the penultimate (PFC) and final foot contact (FFC), FFC peak knee flexion and PFC hip flexion angle (*p* ≤ 0.040, η^2^ = 0.170–0.417). The IG displayed small to large improvements post-intervention in these aforementioned variables (*p* ≤ 0.058, *g* = 0.49–1.21). Turning performance improvements were largely to very largely (*p* ≤ 0.062, *r* or *ρ =* 0.527–0.851) associated with increased mean HPF, more horizontally orientated FFC propulsive force and PFC braking force, and greater pelvic rotation, PFC hip flexion, and PFC velocity reductions. COD speed and technique modification is a simple, effective training strategy that enhances turning performance.

## 1. Introduction

180° turning ability is a fundamental and frequent movement performed in multidirectional sports [1,2,3]. For example, 180° turns are performed in sports such as soccer [4], netball [5], and ultimate Frisbee [6] when transitioning from defence to attack (and vice versa) and to create separation from an opponent, while 180° turning is an integral movement for cricket batsmen to score runs when sprinting and turning between the wickets [7]. Furthermore, change of direction (COD) speed assessments commonly contain 180° turns, including the 505 (traditional and modified) and pro-agility which are typically included in team sports’ testing batteries [8,9]. Importantly, these tests are used for talent identification purposes, such as the National Football League combine [10]. Consequently, training interventions that can enhance an athlete’s mechanical 180° turning ability are of great interest to practitioners working in multidirectional sports.

As changing direction 180° requires athletes to decrease their horizontal momentum to zero [11], rotate their body and plant their foot ahead of their centre of mass (COM) to produce horizontal braking and propulsive impulse, and reaccelerate again, efficient performance of this task is underpinned by the interaction between velocity (linear sprint speed), mechanics (kinetics, kinematics, technique), deceleration (braking strategy), and physical capacity (strength capacity, maximal and rapid force production, neuromuscular control, and muscle activation) [12,13,14,15,16]. As confirmed in a recent meta-analysis [17], plyometric, strength, COD speed training, and combined training methods positively enhance COD speed and manoeuvrability performance containing 180° turns. While these findings appear favourable, it should be noted that most training interventions assess COD performance via completion time using electronic timing gates [17]. This is problematic because the majority of time during a COD speed task is spent linear sprinting and completion times can favour athletes who are linearly fast [8,9]. Therefore, it is uncertain whether improved completion times demonstrated in previous training interventions are attributable to enhanced 180° turning mechanics or simply a product of improved linear speed. As such, there is a paucity of research that has examined the biomechanical mechanisms responsible for improved COD speed performance. Therefore, for a more holistic overview of changes in COD ability, detailed analysis into COD biomechanics such as kinetics, ground contact time (GCT), and velocity of COM are required [8,18].

The biomechanical determinants of 180° turning performance include greater peak and mean horizontal propulsive forces (HPF) in shorter GCTs, more horizontally orientated braking and propulsive forces, and faster approach COM velocities and decreases in velocity over key phases of the COD [11,19,20,21]. Additionally, greater penultimate foot contact (PFC) lower-limb triple flexion [19] and PFC horizontal braking forces (HBF) [15,20,21], and superior medial trunk lean and internal pelvic and foot rotation are related to faster 180° turning performance [19]. Since COD is a skill, coaching and modifying athletes’ 180° turning technique by promoting the aforementioned technical and mechanical characteristics related to faster turning could be an efficacious, simple training method to improve turning performance [15,19,22,23]. This can include increasing PFC hip and knee flexion (COM lowering), increasing PFC HBFs, increasing whole-body rotation, increasing COM velocity profiles, and increasing HPFs over shorter GCTs [15,19,22,23]. For example, 6 weeks’ 180° COD speed and technique modification training which encouraged greater PFC braking, backwards trunk inclination, and a neutral foot position during the final foot contact (FFC) improved completion times and knee joint loads in female netballers [23]. However, it should be noted there was no control group (CG), thus the results should be interpreted cautiously. Furthermore, 6 weeks’ COD speed and technique modification training has been documented to improve 70° side-step cutting performance and movement quality in youth soccer players [22]. Thus, more research is needed which investigates the effects of COD speed and technique modification training on 180° turning biomechanics, while including a CG. 

The aims of this study were three-fold: (1) to examine the effectiveness of a 6 week COD speed and technique modification intervention on 180° turning performance (completion time, GCT, and COM velocity at key instances of COD), whole-body and joint kinetics and kinematics; (2) to establish which biomechanical factors explain changes in performance; and (3) to examine the individual responses (positive/negative/non-responders) following training intervention, since an individual approach has been recommended when analysing the effects of training for a more holistic overview of individual responders [24,25]. It was hypothesised that the COD speed and technique modification programme would improve 180° turning performance, kinetics, and kinematics in multidirectional athletes, and changes in GCT, COM velocity at key instances of the COD, HPFs, and pelvic rotation will be associated with improvements in performance [11,19,20,21]. If COD speed and technique modification does improve 180° turning kinetics, kinematics, and subsequent performance, the results can be used as a potential framework for pitch- or court-based COD speed training programmes. 

## 2. Materials and Methods

### 2.1. Research Design

A non-randomised, controlled 6 week intervention study with a repeated-measures pre-to-post-test design was administered (Figure 1). Male multidirectional sport athletes were recruited for the intervention group (IG) and completed a 6 week COD speed and technique modification training programme (Appendix A). Conversely, male multidirectional sport athletes were the CG, continuing their normal training. Pre-to-post-assessments of 180° turning biomechanics were assessed during a modified 505 (mod505) using 3D motion and ground reaction force (GRF) analysis to evaluate the training intervention’s effectiveness. To control for circadian rhythm, pre-to-post-testing occurred at a similar time of day. 

### 2.2. Subjects

Thirty men from multidirectional sports (amateur/semi-professional) participated in this study. A minimum sample size of 14 per group was calculated from an a priori power analysis using G*Power (Version 3.1, University of Dusseldorf, Düsseldorf, Germany) [26]. This was based on a previously observed effect size of 0.74 (pre-to-post-dependent *t*-test) for completion time changes during 180° turning [23], a power of 0.80, and type 1 error of 0.05. Sixteen men were recruited for the IG (Table 1), while fourteen men who were in the CG continued their normal sport and resistance training sessions (Table 1). Ethical approval was granted for this study (Institutional Ethics Review Board HSR1617-131), and all subjects provided written informed consent to partake in this study. All subjects from both groups had ≥5 years sport-specific training experience, with no history of severe knee injuries prior to testing. All subjects possessed ≥ 1 years’ resistance training experience and performed two resistance training sessions a week as part of a strength mesocycle. All subjects participated in one competitive match and two sport-specific sessions a week. To ensure that no large performance changes were made because of initial conditioning state [22], this study was administered during the competitive season. For study and statistical inclusion, subjects had to complete 10 of the 12 sessions in total (i.e., ≥83% compliance rate) (Table 1). Subsequently, due to match-related injuries, illness, and methodological issues during post-testing, three and two subjects withdrew from the IG and CG, respectively. This resulted in sample sizes of 13 and 12 (Figure 1, Table 1), respectively. 

### 2.3. Procedures 

Warm up, mod505, marker placement, and 3D motion analysis procedures were based on previously published methodologies [11,19,27]. Briefly, each subject performed six trials of a mod505 (5 m entry and 5 m exit) (right limb turn) as quickly as possible and was provided with standardised footwear to control for shoe–surface interface (Balance W490, New Balance, Boston, MA, USA). Completion time was assessed using a pair of Brower timing lights placed at hip height at the start (finish) line (Draper, UT, USA), and subjects adopted a two-point staggered stance, 0.5 m behind the start line. Testing took place on an indoor track (Mondo, SportsFlex, 10 mm; Mondo America Inc., Mondo, Summit, NJ, USA). Marker and force data were collected over the PFC and FFC using ten Qualisys Oqus 7 (Gothenburg, Sweden) infrared cameras (240 Hz) operating through Qualisys Track Manager software (Qualisys, version 2.16 (Build 3520), Gothenburg, Sweden) and GRFs were collected from two 600 mm × 900 mm AMTI (Advanced Mechanical Technology, Inc., Watertown, MA, USA) force platforms (Model number: 600900) embedded into the running track sampling at 1200 Hz. Using the pipeline function in visual 3D, joint coordinate (marker) and force data were smoothed using a Butterworth low-pass digital filter with cut-off frequencies of 15 and 25 Hz, respectively. The kinematic model process was based on previously reported methodologies [11,19,27]. Joint kinematics and GRFs were also calculated using Visual 3D (C-motion, version 6.01.12, Germantown, MD, USA), while GRF braking characteristics were normalised to body weight, with vertical, anterior–posterior, and medio–lateral corresponding to Fz, Fx, and Fy, respectively. Horizontal COM velocity profiles over key instances of the COD were calculated as described previously [11,19]. 

#### 2.3.1. Turning Biomechanical Variables

Appendix A provides a full description of the variables examined, definitions, and calculations. In summary, the following biomechanical variables were examined because they are determinants of 180° turning performance [11,19,20,21]: GCT, approach velocity (PFC touch-down), change in PFC velocity, velocity at FFC (touch-down) and exit velocity (FFC toe-off); mean HPF, PFC horizontal to vertical braking force ratio and FFC horizontal to vertical propulsive force ratio, lateral trunk flexion angle, FFC trunk inclination angle, pelvis rotation angle, initial foot progression angle (IFPA), FFC knee flexion angle range of motion (ROM), and PFC peak hip flexion (HFA) and knee flexion angle (KFA). Completion time, GCT, and exit velocity were the considered primary outcome variables. Five trials were used in the analysis of each subject and the average of individual trial peaks for each variable were calculated [27]. A subset of the sample (*n* = 10) performed the COD task on two separate occasions (7 days apart) to determine between-session reliability, and the COD biomechanical data were considered highly reliable (intraclass correlation coefficient = 0.728–0.952, coefficient of variation = 2.8–13.9%).

#### 2.3.2. The 6 Week COD Speed and Technique Modification Training Intervention

Appendix A provides a detailed overview of the 6 week COD technique modification intervention. The IGs performed this twice a week (30 min per session, ≥48 h between sessions) on match day +2/+3 (48–72 h post-match) and match day −2 (48 h pre-match). The intervention was adapted from a previously successful 6 week COD speed and technical modification training intervention [22], which focused on pre-planned low intensity decelerations, cuts, and turns (weeks 1–2), before progressing intensity via velocity and angle (weeks 3–4), and introducing a stimulus with increased intensity (weeks 3–6) to increase complexity, target technique integrity (reinforce optimal technique), and to provide variation within the COD development framework [1,2,3]. The duration, distances, and number of CODs performed each session were similar to previous research [22,23]. The principle researcher delivered all sessions who is a qualified strength and conditioning coach (certified strength and conditioning specialist), and they took place in the same facility and surface as used for 3D testing. Athlete-to-coach ratios ranged from 5:1 to 8:1. The technical modifications focused on three aspects and cues [3,15,22,23]: (1) “slam on the brakes and drop the hips” (to increase PFC triple flexion and PFC HBF, and reduce horizontal momentum); (2) “cushion and push/punch the ground away” (to enhance HPF, encourage active limb at touch-down and impulsive push-off, and facilitate a shorter GCT and rapid transition from braking to propulsion); and (3) “face towards the direction of travel” (to encourage whole-body rotation to reduce redirection demands). Subjects were provided with individualised technical feedback between repetitions, and to permit better motor skill retention, externally directed verbal coaching cues were used [3,22].

### 2.4. Statistical Analyses

All statistical analyses were performed using SPSS v25 (SPSS Inc., Chicago, IL, USA) and Microsoft Excel (version 2016, Microsoft Corp., Redmond, WA, USA). A Shapiro–Wilks test was used to examine normality for all variables. A two-way mixed analysis of variance (ANOVA) (group; time) with group as a between-participants factor measured at 2 levels (IG and CG), and time (pre- and post-training measures) the within-subject factor. This was used to identify any significant interaction (group × time) effects for outcome variables between IG and CG, pre-to-post-testing. A Bonferroni-corrected pairwise comparison design was used to further analyse the effect of the group when a significant interaction effect was observed. Partial eta squared (η^2^) effect sizes were calculated for all ANOVAs and interpreted as small (0.010–0.059), medium (0.060–0.149), and large (≥0.150) [28].

A paired-samples *t*-tests (parametric) or Wilcoxon-sign ranked tests (non-parametric) was used to examine pre-to-post-changes in variables. Magnitudes of differences were measured using Hedges’ *g* effect sizes with 95% confidence intervals (CI), and interpreted using Hopkins’ scale [29]. Group mean changes were calculated and interpreted as ratios relative to the smallest worthwhile change (SWC). The SWC was calculated as 0.2 × between-subject SD [25]. Outcome variables and changes in outcome variables between the IG and CG were assessed using independent sample *t*-tests or Mann–Whitney U tests, with effect sizes as outlined above. Furthermore, to connect changes in turning performance with biomechanical changes (IG pre-to-post-mean change data), Pearson’s correlations or Spearman’s correlations were calculated with 95% CIs, and *p* values Bonferroni corrected to control for type 1 error. Correlations were interpreted using Hopkins’ scale [30], and a cut-off value of ≥0.40 was considered relevant [31]. An alpha level of *p* ≤ 0.05 was used to define statistical significance for all tests. Finally, similar to previous work [25], individual analyses were performed to quantify for each variable and each group the number of positive, negative, and non-responders. For all variables of interest, positive or negative responses were considered as an individual change ≥SWC, while trivial responses (non-responder) was considered ≤SWC. 

## 3. Results

Results of the two-way mixed ANOVAs are presented in Table 2 and IG and CG pre-to-post-changes in turning biomechanics are presented in Table 3 containing descriptive statistics, *p* values, effect sizes, mean differences, and individual responses. A large, significant interaction effect for completion time was observed (Table 2), with the IG showing significantly shorter completion times (*p* = 0.029, *g* = −0.90 ± 0.82) and GCTs (*p* = 0.047, *g* = −0.80 ± 0.82) post-intervention compared to the CG (Table 3). A moderate improvement in completion time was observed for the IG post-intervention only (Table 3), and the mean changes were largely greater than the CGs (*p* < 0.001, *g* = −1.57 ± 0.90). No significant interaction effects or pre-to-post-improvements were observed for the other performance variables, but individual variation was observed (Table 2 and Table 3).

Large, significant interaction effects were observed for mean HPF, FFC and PFC mean horizontal to vertical braking and propulsive force ratios (Table 2), respectively, with the IG showing greater mean HPFs (*p* = 0.005, *g* = −1.21 ± 0.85), FFC (*p* = 0.010, *g* = 1.08 ± 0.84) and PFC mean horizontal to vertical propulsive and braking (*p* = 0.124, *g* = 0.63 ± 0.80) force ratios post-intervention compared to the CG (Table 3), respectively. Moderately to largely improved mean HPFs, FFC and PFC mean horizontal to vertical propulsive and braking force ratios were observed for the IG post-intervention only (Table 3), and these mean changes were moderately to largely greater than the CGs (*p* ≤ 0.023, *g* = 0.90–1.21 ± 0.82–0.86). No significant interaction effects or pre-to-post-improvements were observed for the other GRF variables, but individual variation was observed (Table 2 and Table 3). Large significant interaction effects were observed for FFC KFA ROM and PFC peak HFA (Table 2), with the IG showing lower FFC KFA ROM (*p* = 0.002, *g* = 1.31 ± 0.86) and greater PFC peak HFAs (*p* = 0.297, *g* = 0.41 ± 0.79) post-intervention compared to the CG (Table 3). Small and moderate improvements in FFC KFA ROM and PFC peak HFA post-intervention were observed (Table 3), respectively, and these mean changes were moderately greater than the CGs (*p* ≤ 0.040, *g* = −0.84–0.93 ± 0.79–0.83). No other significant interaction effects for the other kinematic variables were observed (Table 2); however, small to moderate increases in FFC trunk inclination angle, pelvic rotation, and IFPA post-intervention were observed for the IG (Table 3). 

Completion time improvements were largely associated with greater PFC peak HFA and greater PFC horizontal to vertical mean braking force ratios (Table 4). Additionally, GCT improvements were very largely associated with greater FFC horizontal to vertical mean propulsive force ratios and greater pelvic rotation; and largely associated with greater mean HPF, greater PFC velocity reductions, and greater PFC peak HFA (Table 4). Finally, exit velocity improvements were very largely associated with greater mean HPFs and largely associated with greater FFC horizontal to vertical mean propulsive force ratios (Table 4).

## 4. Discussion

This study’s aims were three-fold: (1) to examine the effectiveness of a 6 week COD speed and technique modification intervention on 180° turning performance, whole-body and joint kinetics and kinematics; (2) to establish which biomechanical factors explain changes in performance; and (3) to examine the individual responses following the intervention. The primary findings were that COD speed and technique modification resulted in meaningful improvements in turning performance (completion time and GCT), kinetics (HPF and PFC braking and FFC propulsive force orientation) and mechanics (pelvic rotation, FFC KFA ROM, and peak PFC HFA) which were meaningfully better than the CG (Table 2 and Table 3, Figure 2 and Figure 3), supporting the study hypotheses and previously successful 6 week COD speed technique modification interventions [22,23]. The positive improvements in turning performance were primarily attributable to increases in mean HPF, more horizontally orientated FFC propulsive force and PFC braking force, greater pelvic rotation, greater PFC HFA, and greater PFC velocity reductions (Table 4), which are also in line with the study hypotheses. Consequently, 6 weeks’ COD speed and technique modification is an effective training modality for improving 180° turning performance, kinetics and kinematics, which practitioners can simply and easily administer on the pitch or court.

While it is well established that COD speed training enhances completion time [17,22,23], the biomechanical mechanisms which explain improved COD performance are relatively unknown. To the best of our knowledge, Jones et al. [23] is the only other study to investigate the biomechanical effects of COD speed and 180° turning technique modification, reporting improvements in completion time and reduced initial foot progression angles, knee abductions moments, and trunk inclination angle, and no significant changes in approach velocity. However, the results should be interpreted cautiously because there was no CG, the population were not highly trained, and the programme focused on injury mitigation. The present study overcame these limitations by comprehensively examining the biomechanical effects of COD speed and 180° turning technique modification with external verbal cues while including a CG and utilising a population with greater training experience. Corroborating previous work [23], this modality was highly effective in enhancing mod505 completion times, and producing meaningful improvements in turning GCTs, HPFs, technical orientation of PFC braking force and FFC propulsive force, and mechanics including pelvic rotation, PFC HFA, and FFC knee flexion ROM (Table 2 and Table 3, Figure 2 and Figure 3); which have all been highlighted as determinants of faster 180° COD performance [11,19,20,21]. External cueing has been strongly recommended for speed development [32], motor skill retention [3,32], while a cutting technique modification which utilised external cues was also effective in improving cutting completion times, COD deficits, and movement quality in youth soccer players [22]. Practitioners are therefore encouraged to use external verbal cues when implementing COD speed training to promote favourable changes in turning performance, kinetics, and kinematics. 

Researchers have attempted to quantify kinetic or kinematic changes during cutting following a training intervention [33,34,35], but unfortunately did not holistically assess performance (i.e., completion time, GCT, and COM velocity profiles). Nevertheless, improved cutting GRF magnitudes in shorter durations have been demonstrated following 10 weeks’ eccentric overload training [35] and 12 weeks’ neuromuscular training [34], while a multicomponent inter-segmental control training intervention produced favourable changes in cutting kinematics and kinetics linked to faster performance [33]. Uniquely, in the present study, positive improvements in IG mean HPF and horizontal to vertical propulsive force ratios were observed post-intervention (Table 2 and Table 3, Figure 2 and Figure 3), and changes in these variables were largely to very largely associated with improvements in exit velocity (Table 4). Greater horizontal propulsive forces have previously been associated with faster mod505 performance [19,20,21], attributed to greater force magnitudes increasing impulse which based on the impulse-momentum relationship leads to greater changes in velocity [3,20,21]. Additionally, changes in acceleration are proportionate to the direction force is applied; thus, a more horizontally orientated propulsive force vector should facilitate effective net acceleration [19,36]. This positive adaptation could be attributed to the cue to “push or punch the ground away” which practitioners should therefore consider implementing when coaching 180° turns. 

Greater PFC HFAs and PFC horizontal to vertical mean braking force ratios were demonstrated post-intervention by the IG and associated with changes in completion time (Table 3 and Table 4, Figure 3). Greater PFC HFAs have been associated with faster mod505 completion times [19], contributing to a lower COM for effective FFC weight acceptance [15,19], while a more horizontally orientated braking force vector helps facilitate more effective net deceleration [15,19]; a key aspect for successful 180° turning [11,19]. Similarly, greater velocity reductions over the PFC were largely associated with shorter FFCs GCTs (Table 4). Collectively, the abovementioned positive modifications highlight the multistep nature of COD and the importance of PFC braking for effective 180° turning [11,15,19]. The longitudinal coaching and cueing to “slam on the brakes and drop the hips” during the intervention most likely explain the improvements in PFC braking characteristics. Moreover, moderate increases in IG pelvic rotation were demonstrated post-intervention (Table 3), and changes in pelvis rotation were very largely associated with shorter GCTs (Table 4). Encouraging greater pelvic rotation reduces the redirection requirements during the push-off [15,19], and improvements in this technique are likely attributable to the cue “face towards the direction of travel”. Consequently, the present study highlights the effectiveness of coaching with external cueing and technical feedback for promoting favourable changes in turning kinetics and kinematics, while providing further insight into the biomechanical mechanisms responsible for enhanced 180° turning performance. 

Most training intervention studies adopt a within- and between-group analysis approach, typically evaluating the success of an intervention based on group means [17]. Uniquely, the present study monitored the individual responses to intervention, whereby this approach has been increasing in practice [24,25]. Table 3 and Figure 2 and Figure 3 illustrate the IGs individual responses, indicating a substantial proportion (≥77%) demonstrated positive changes greater than the SWC for completion time, technical orientation of PFC braking and FFC propulsive force and PFC HFA. Additionally, 69% of the IG displayed smaller KFA ROMs which is important because reduced knee flexion have been linked to shorter GCTs and superior exit velocities during cutting [37]. Although no significant interaction effects or pre-to-post-improvements were observed for key metrics such as mean HPF, approach and exit velocity (Table 2), though changes were small to moderate (Table 3), individual analysis showed that 54–62% of the IG demonstrated positive improvements in approach and exit velocity, and mean HPF (Table 3, Figure 2), underlining the importance of an individual approach when evaluating training intervention effectiveness. No negative responders were observed for completion time, while 15% and 23% of the IG responded negatively with respect to GCTs and exit velocity (Table 3, Figure 2), respectively. Although not examined in the present study, speculatively the negative or non-responders may not have had the physical capacity to adopt the favourable mechanics associated with faster performance [11,18,38] and may have warranted some pre-conditioning to supplement the technique modification intervention [15]. Alternatively, they may have responded more positively to an alternative training modality (i.e., strength or plyometric training), but further research is required to support this contention. 

It is worth noting that a pre-planned 180° turn from a right-limb push off was only examined in the present study; therefore, cautious extrapolation to CODs of different angles and unplanned CODs is recommended. Additionally, a 6 week intervention was administered which was in line with previous COD speed and technique modification interventions [22,23,39]; however, it is unclear whether 6 weeks is an optimal duration, and it is unknown whether longer training periods are necessary to elicit greater improvements. Despite this, the present study demonstrated meaningful short-term improvements in turning performance in just 6 weeks, and could therefore be a worthwhile modality for practitioners in multidirectional sports. Further research is needed which explores the optimal dosages and minimum effective dose to elicit improvements in turning performance, kinetics, and kinematics. Moreover, as no continued follow-up testing (retention testing) was performed, it is uncertain whether and how long improvements in turning technique and performance can be maintained for. Consequently, greater insights into the short- and long-term retention of modified turning technique warrants further investigation. Nevertheless, COD provides the mechanical and physical basis for agility [40]; therefore, enhancements in the mechanics and physical ability to COD (i.e., fast mover) should equate to partial transfer to improved agility [1,2,40]. Lastly, athletes do change direction and sprint to pre-determined locations on a court or pitch [40], and COD speed and manoeuvrability tasks which contain 180° turns are included in team sports’ testing batteries [8,9], which are used for talent identification purposes, such as the National Football League combine [10]. Therefore, the findings of this study have important implications for the development of effective 180° turning training programmes and coaching of 180° turning technique. 

## 5. Conclusions

This is the first study to examine the biomechanical effects of COD speed and technique modification on 180° turning performance, while including a CG, demonstrating meaningful improvements in turning performance, kinetics, and kinematics. Specifically, athletes following the intervention were generally able to produce greater mean HPF magnitudes in shorter GCTs, apply and orientate PFC braking force and FFC propulsive force more horizontally, display greater pelvis rotation, smaller FFC knee flexion ROM, and greater PFC velocity reduction and peak HFAs, which cumulatively resulted in superior turning performance. Consequently, 6 weeks’ COD speed and technique modification training with externally directed verbal coaching cueing (“slam on the bakes and drop the hips, cushion and push/punch the ground away, and face towards the direction of travel”) and technical feedback is an effective training modality that can enhance turning performance, which practitioners can simply and easily incorporate into their pitch- or court-based training programmes.

## Figures and Tables

**Figure 1 sports-09-00073-f001:**
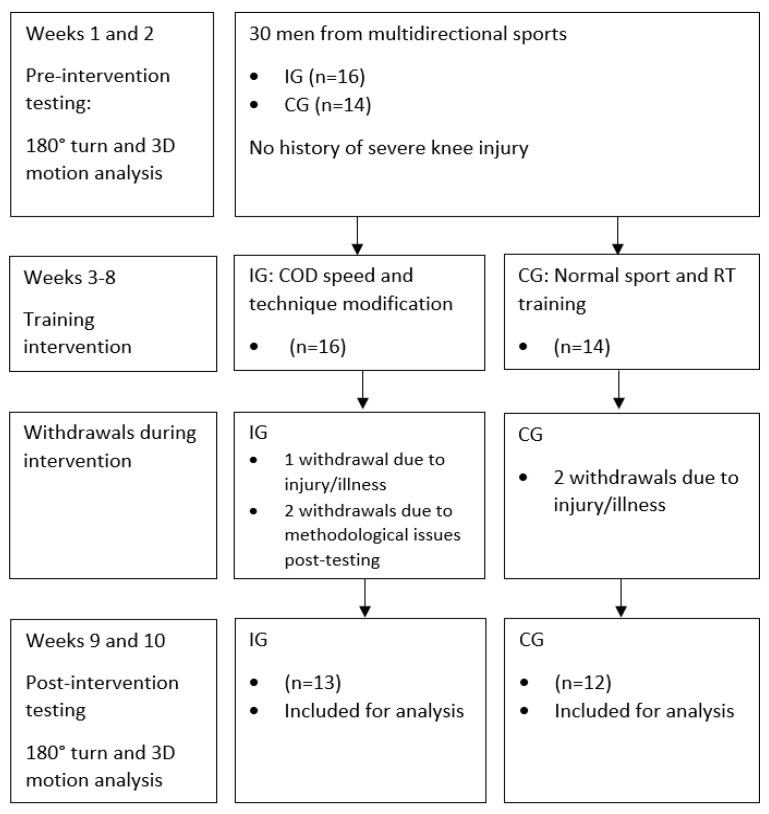
Flow diagram of subject participation throughout all stages of the intervention study. IG: intervention group; CG: control group; COD: change of direction; 3D: three dimensional.

**Figure 2 sports-09-00073-f002:**
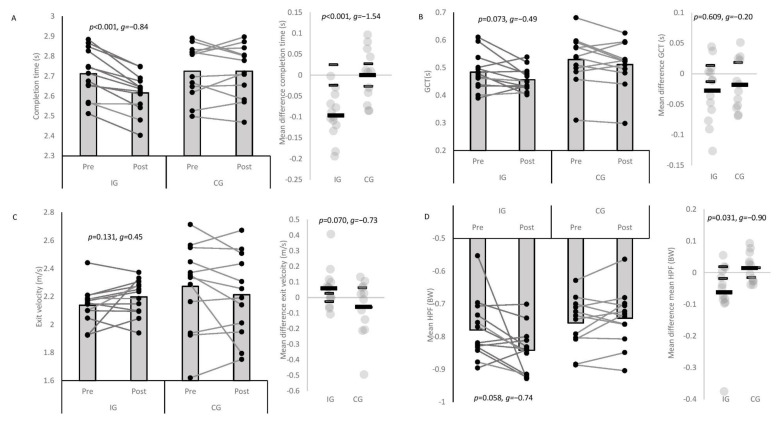
Pre-to-post-changes in mod505 performance variables with individual plots (connected lines). (**A**) Completion time; (**B**) GCT; (**C**) exit velocity; (**D**) mean HPF; IG: intervention group; CG: control group; GCT: ground contact time; HPF: horizontal propulsive force; grey column bars denote mean. Black rectangle denotes mean difference (pre-to-post-change). Smaller grey rectangles indicate smallest worthwhile change; thus, individual changes which exceed these are considered a positive or negative responder, and changes less than the smallest worthwhile change indicate a non-responder.

**Figure 3 sports-09-00073-f003:**
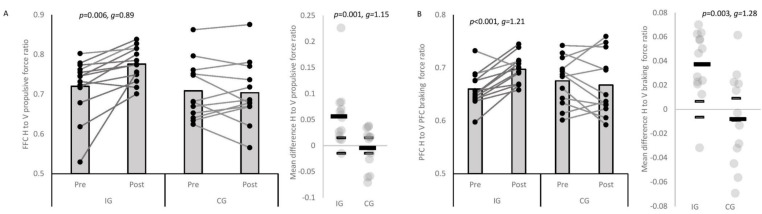
Pre-to-post-changes in mod505 ground reaction force variables with individual plots (connected lines). (**A**) FFC H to V propulsive force ratios; (**B**) PFC H to V braking force ratios; IG: intervention group; CG: control group; H: horizontal; V: vertical; grey column bars denote mean. Black rectangle denotes mean difference (pre-to-post-change). Smaller grey rectangles indicate smallest worthwhile change; thus, individual changes which exceed these are considered a positive or negative responder, and changes less than the smallest worthwhile change indicate a non-responder.

**Table 1 sports-09-00073-t001:** Subject characteristics for the intervention and control group.

	Intervention Group	Control Group
Age (years)	23.5 ± 5.2	22.2 ± 5.0
Height (m)	1.80 ± 0.05	1.76 ± 0.08
Mass (kg)	81.6 ± 11.4	72.7 ± 12.4
Sporting population	soccer *n* = 12, rugby *n* = 4	soccer *n* = 9, rugby *n* = 4, field hockey *n* = 1
Compliance rate	11.9 ± 0.3 sessions (99.4 ± 2.3%). 12 subjects completed 12 (100%) sessions and one subject completed 11 sessions (91.7%)	-

**Table 2 sports-09-00073-t002:** Two-way mixed ANOVAs for 180° turning performance, kinetic and kinematic variables.

Variable	Group (Interaction)		
	*p* Value	η^2^	Power
Completion time	<0.001 **	0.417	0.973
GCT	0.615	0.011	0.078
Approach velocity	0.706	0.006	0.066
PFC change in velocity	0.465	0.023	0.110
Velocity at FFC	0.741	0.005	0.062
Exit velocity	0.071	0.135	0.443
FFC mean HPF	0.031 *	0.186	0.594
FFC H to V mean propulsive ratio	0.008 *	0.271	0.800
PFC mean HBF	0.929	0.000	0.051
PFC H to V mean braking force ratio	0.004 *	0.314	0.874
Lateral trunk flexion	0.758	0.004	0.060
FFC trunk inclination	0.802	0.003	0.057
Pelvic rotation	0.381	0.034	0.137
IFPA	0.458	0.024	0.112
FFC KFA ROM	0.025 *	0.200	0.633
PFC peak HFA	0.040 *	0.170	0.548
PFC peak KFA	0.720	0.006	0.064
Trivial η^2^ (<0.010)	Small η^2^ (0.010–0.059)	Medium η^2^ (0.060–0.149)	Large η^2^ (≥0.150)

Key: GCT: ground contact time; FFC: final foot contact; PFC: penultimate foot contact; HPF: horizontal propulsive force; HBF: horizontal braking force; H: horizontal; V: vertical; KFA: knee flexion angle; ROM: range of motion; HFA: hip flexion angle; IFPA: initial foot progression angle; *: *p* ≤ 0.05; **: *p* ≤ 0.001.

**Table 3 sports-09-00073-t003:** The 180° turn pre-to-post changes in performance, kinetics, and kinematic variables for IG and CG.

Group	Variable	Pre		Post			Hedges’ *g*		Pre-Post Mean Difference		SWC	Ratio to SWC	Individual Responses
		Mean	SD	Mean	SD	*p*	*g*	±CI	Mean	SD			(Positive, Non, Negative)
IG	Completion time (s)	2.712	0.122	2.616 ^b^	0.099	<0.001 ^**^	−0.84	0.80	−0.096c	0.055	0.024	3.9	12-1-0
	GCT (s)	0.484	0.067	0.456 ^b^	0.041	0.073	−0.49	0.78	−0.028	0.051	0.013	2.1	6-5-2
	Approach velocity (m/s)	3.96	0.33	4.00	0.24	0.625	0.13	0.77	0.04	0.29	0.07	0.6	8-1-4
	PFC change in velocity (m/s)	−1.30	0.35	−1.40	0.22	0.25	−0.33	0.77	−0.10	0.29	0.07	1.4	6-4-3
	Velocity at FFC (m/s)	2.65	0.21	2.60	0.26	0.374	−0.24	0.77	−0.06	0.23	0.04	1.4	5-3-5
	Exit velocity (m/s)	2.14	0.13	2.20	0.13	0.131	0.45	0.78	0.06	0.13	0.03	2.3	7-3-3
CG	Completion time (s)	2.725	0.134	2.724	0.133	0.996	0.00	0.80	0.000	0.063	0.027	0.0	5-3-4
	GCT (s)	0.529	0.092	0.511	0.085	0.157	−0.20	0.80	−0.018	0.041	0.018	1.0	6-2-4
	Approach velocity (m/s)	4.09	0.34	4.09	0.30	0.978	0.01	0.80	0.00	0.20	0.07	0.0	3-3-6
	PFC change in velocity (m/s)	−1.38	0.20	−1.42	0.20	0.332	−0.15	0.80	−0.03	0.11	0.04	0.8	5-4-3
	Velocity at FFC (m/s)	2.70	0.34	2.67	0.28	0.600	−0.09	0.80	−0.03	0.19	0.07	0.4	7-1-4
	Exit velocity (m/s)	2.27	0.31	2.21	0.30	0.276	−0.19	0.80	−0.06	0.18	0.06	0.9	4-3-5
IG	FFC mean HPF (BW)	−0.78	0.09	−0.84 ^c^	0.07	0.058	−0.74	0.79	−0.06 ^b^	0.11	0.02	3.4	8-3-2
	FFC H to V mean propulsive ratio	0.72	0.07	0.78 ^b^	0.04	0.006 ^*^	0.89	0.81	0.06 ^b^	0.06	0.01	3.8	10-3-0
	PFC mean HBF (BW)	−0.51	0.06	−0.53	0.06	0.087	−0.34	0.77	−0.02	0.04	0.01	1.8	7-2-4
	PFC H to V mean braking force ratio	0.66	0.03	0.70 ^c^	0.03	<0.001 ^**^	1.21	0.84	0.04 ^c^	0.03	0.01	5.7	12-0-1
CG	FFC mean HPF (BW)	−0.76	0.08	−0.74	0.09	0.299	0.17	0.80	0.01	0.05	0.02	0.9	5-2-5
	FFC H to V mean propulsive ratio	0.71	0.07	0.70	0.08	0.712	−0.06	0.80	0.00	0.04	0.01	0.3	6-3-3
	PFC mean HBF (BW)	−0.52	0.07	−0.54	0.10	0.174	−0.23	0.80	−0.02	0.05	0.01	1.4	9-1-2
	PFC H to V mean braking force ratio	0.68	0.05	0.67	0.06	0.558	−0.15	0.80	−0.01	0.04	0.01	0.9	5-2-5
IG	Lateral trunk flexion (°)	3.9	4.5	4.9	7.5	0.602	0.15	0.77	1.0	6.6	0.9	1.1	6-4-3
	FFC trunk inclination (°)	36.4	12.9	41.6	10.4	0.075	0.43	0.78	5.2	9.6	2.6	2.0	7-3-3
	Pelvic rotation (°)	76.9	10.0	83.2	9.0	0.061	0.64	0.79	6.3	11.0	2.0	3.1	7-3-3
	IFPA (°)	48.2	16.4	52.1	14.5	0.384	0.24	0.77	3.9	15.6	3.3	1.2	6-2-5
	FFC KFA ROM (°)	44.5	5.5	42.1 ^c^	4.1	0.012 ^*^	−0.49	0.78	−2.4 ^b^	3.0	1.1	2.2	9-2-2
	PFC peak HFA (°)	88.8	11.9	95.6	7.8	0.003 ^*^	−0.65	0.79	−6.8 ^b^	6.5	2.4	2.9	10-2-1
	PFC peak KFA (°)	113.0	6.9	114.0	6.6	0.458	0.14	0.77	1.0	4.5	1.4	0.7	6-0-7
CG	Lateral trunk flexion (°)	10.9	11.7	12.6	10.2	0.224	0.15	0.80	1.7	4.5	2.3	0.7	5-6-1
	FFC trunk inclination (°)	32.1	13.3	38.8	12.9	0.248	0.50	0.81	6.7	19.1	2.7	2.5	7-3-2
	Pelvic rotation (°)	89.2	24.9	92.2	22.0	0.140	0.12	0.80	3.0	6.6	5.0	0.6	3-8-1
	IFPA (°)	60.7	22.4	69.3	12.9	0.082	0.45	0.81	8.6	15.5	4.5	1.9	7-4-1
	FFC KFA ROM (°)	49.1	5.8	49.8	7.0	0.502	0.11	0.80	0.7	3.6	1.2	0.6	4-4-4
	PFC peak HFA (°)	89.5	18.7	91.1	19.9	0.337	−0.08	0.80	−1.6	5.4	3.7	0.4	3-6-3
	PFC peak KFA (°)	110.1	9.5	110.3	10.3	0.920	0.02	0.80	0.2	6.2	1.9	0.1	5-2-5
Trivial ES (≤0.19)		Small ES (0.20–0.59)				Moderate ES (0.60–1.19)				Large ES (1.20–1.99)			

Key: GCT: Ground contact time; FFC: Final foot contact; PFC: Penultimate foot contact; HPF: Horizontal propulsive force; HBF: Horizontal braking force; H: Horizontal; V: Vertical; KFA: Knee flexion angle; ROM: Range of motion; HFA: Hip flexion angle; IFPA: Initial foot progression angle; SD: Standard deviation; SWC: Smallest worthwhile change; IG: Intervention group; CG: Control group; CI: 95% Confidence interval; ES: Effect size; *: *p* ≤ 0.05; **: *p* ≤ 0.001; ^b^: significantly different (*p* < 0.05) from CG with moderate effect size; ^c^: significantly different (*p* < 0.05) from CG with large effect size.

**Table 4 sports-09-00073-t004:** The 180° change of direction associations between changes in performance variables with changes in technical and mechanical variables.

Performance Variable	Associated with	Correlation Value with 95% Confidence Interval	Descriptor
Improvements in completion time	Greater PFC peak hip flexion Greater PFC H to V mean braking force ratio	*ρ* = 0.654 ± 0.362, *p* = 0.009 *ρ* = −0.585 ± 0.404, *p* = 0.036	Large Large
Improvements in GCT	Greater FFC H to V mean propulsive force ratio Greater pelvis rotation Greater mean HPF Greater Δ PFC velocity Greater PFC peak hip flexion	*r* = −0.851 ± 0.195, *p* < 0.001 *r* = −0.728 ± 0.309, *p* = 0.005 *r* = 0.646 ± 0.368, *p* = 0.017 *r* = −0.531 ± 0.433, *p* = 0.062 *ρ* = 0.527 ± 0.435, *p* = 0.026	Very large Very large Large Large Large
Improvements in exit velocity	Greater mean HPF Greater FFC H to V mean propulsive force	*r* = −0.712 ± 0.321, *p* = 0.006 *r* = 0.628 ± 0.379, *p* = 0.022	Very large Large

Key: FFC: final foot contact; PFC: penultimate foot contact; COD: change of direction; H: horizontal; V: vertical; HPF: horizontal propulsive force.

## Data Availability

The datasets used and/or analysed during the current study are available from the corresponding author on reasonable request.

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
