# Peer review of "Change of Direction Speed and Technique Modification Training Improves 180° Turning Performance, Kinetics, and Kinematics"

_sports, 2021, doi:10.3390/sports9060073_

Round 1

Reviewer 1 Report

General comments:

The manuscript entitled Change of Direction Speed and Technique Modification Training Improves 180° Turning Performance, Kinetics, and Kinematics is an experimental study investigating the influence of proposed training program on the ability to perform 180° turns in semi-professional athletes. The research involve proper design and methodology. Presentation of the individual changes significantly increases the attractiveness of the results section. Moreover, current study provide practical information for coaches who are trying to improve the agility of their players. Therefore, I suggest to accept the paper after the minor corrections mentioned below.

Specific comments:

Introduction

I have noticed that authors tend to use extended sentences. For example in line 68 we can find a 7-line long sentence that includes 7 abbreviations and another few explanations in parentheses. Although, I totally agree with the content of this section, I would like to encourage the authors to split these statements into shorter sentences. It will improve the reading flow undoubtedly.

When presenting the aim of the study authors stated: “The aims of the study were two-fold:” 1) evaluate the effectiveness of the training; and 2) identify the factors…

While below we can find the sentence that “… a secondary aim was to examine the individual responses…” So there were actually 3 aims of the research. The number of study purposes is incoherent. Please notice, that at the beginning of the Discussion section aims of the research are mentioned as well.

Materials and Methods

In general, the methodology is described very well, however some additional information should be added.

In which days of the microcycle the experimental sessions were performed (how many days after the match and/or before the match? This is important detail for coaches who would like to apply presented training program.

I couldn’t find the information about the surface during the tests. Was it a running track in the laboratory? It need to stated clearly.

Line 159: “they” instead of “hey”?

Line 202: Please insert an abbreviation of partial eta squared in the statistical analyses description.

Results

Presentation the individual changes improves the quality of the paper. Well done!

Author Response

Reviewer 1

The manuscript entitled Change of Direction Speed and Technique Modification Training Improves 180° Turning Performance, Kinetics, and Kinematics is an experimental study investigating the influence of proposed training program on the ability to perform 180° turns in semi-professional athletes. The research involve proper design and methodology. Presentation of the individual changes significantly increases the attractiveness of the results section. Moreover, current study provide practical information for coaches who are trying to improve the agility of their players. Therefore, I suggest to accept the paper after the minor corrections mentioned below.

Specific comments:

Introduction

I have noticed that authors tend to use extended sentences. For example in line 68 we can find a 7-line long sentence that includes 7 abbreviations and another few explanations in parentheses. Although, I totally agree with the content of this section, I would like to encourage the authors to split these statements into shorter sentences. It will improve the reading flow undoubtedly.

Response: Thank you for your comment. Apologies for the long sentence. We have rearranged the sentence and divided into two sentences. Please see line 70-76.

When presenting the aim of the study authors stated: “The aims of the study were two-fold:” 1) evaluate the effectiveness of the training; and 2) identify the factors…

While below we can find the sentence that “… a secondary aim was to examine the individual responses…” So there were actually 3 aims of the research. The number of study purposes is incoherent. Please notice, that at the beginning of the Discussion section aims of the research are mentioned as well.

Response: Thank you for your comment and suggestion. We have now amended this to three aims, and we have amended the introduction and discussion accordingly.  Please see line 86, 92, and 307.

Materials and Methods

In general, the methodology is described very well, however some additional information should be added.

In which days of the microcycle the experimental sessions were performed (how many days after the match and/or before the match? This is important detail for coaches who would like to apply presented training program.

Response: Thank you for your comment and identifying this. We have now included this. We have added the following “A six-week COD technique modification intervention described in Supplementary materi-al 1, was performed by the IG twice a week (30 minutes per session, ≥48 hours between sessions) on match day +2/+3 (48-72 hours post-match) and match day -2 (48 hours pre-match).” Please see line 196.

I couldn’t find the information about the surface during the tests. Was it a running track in the laboratory? It need to stated clearly.

Response: Thank you for your comment and identifying this. We have now included this. Please see line 161.

Line 159: “they” instead of “hey”?

Response: Thank you for your comment and identifying this. This has been amended on line 180.

Line 202: Please insert an abbreviation of partial eta squared in the statistical analyses description.

Response: Thank you for your comment and identifying this. This has been added on line 224.

Results

Presentation the individual changes improves the quality of the paper. Well done!

Response: Thank you for your comment and we are glad that you think they are insightful.

Reviewer 2 Report

Authors presents very interesting reseach about  kinetics, kinematics,  of the training performance for which the results can be used to assist in the development of more effective. In conclusion is suggested that technical feedback is an effective training modality that can enhance the effects of training programmes. The analysis was realised for high number of cases and conclusions have not any mistakes. I suggest this article to publication.

Apart of them in my opinion the figures 2, 3 are very complicated and should be better described for more understading presented data and analyse. What the lines meaning?

Author Response

Reviewer 2

Authors presents very interesting reseach about  kinetics, kinematics,  of the training performance for which the results can be used to assist in the development of more effective. In conclusion is suggested that technical feedback is an effective training modality that can enhance the effects of training programmes. The analysis was realised for high number of cases and conclusions have not any mistakes. I suggest this article to publication.

Apart of them in my opinion the figures 2, 3 are very complicated and should be better described for more understading presented data and analyse. What the lines meaning?

Response: Thank you for your comments. The figure legends have been amended with an explanation and more detailed description provided. Please see line 299-301 and 303-305.  

Reviewer 3 Report

The paper is well written and clear, the statistical analysis is very thorough and detailed; figures and tables are clear and easy readable; both the structure of the text and the references list follow the guide line.

In my opinion the athors have done a good job and the paper is ready for the publicaton in the present form

Author Response

Reviewer 3

The paper is well written and clear, the statistical analysis is very thorough and detailed; figures and tables are clear and easy readable; both the structure of the text and the references list follow the guide line.

In this paper the authors want to evaluate the effects of change of direction speed and technique modification training on 180 ° turning performance from a complete biomechanical point of view; the aim is new and interesting to deepen a complex field as sport is. The topic is original and the basis of this research is well clarified in the introduction. The overall merit of the authors is to describe a specific task widely used across various sports in a very specific way, following all the good practice in terms of both the recruitment of patient and statistical analysis. The paper is well written, clear and easy to read not only in terms of text but also figures and tables. All the paper is well structured and the aims presented in the introduction are reported in the discussion with order and clarity. 

In my opinion the athors have done a good job and the paper is ready for the publicaton in the present form

Response: Thank you for your kind words